# Gibberellin Positively Regulates Tomato Resistance to *Tomato Yellow Leaf Curl Virus* (TYLCV)

**DOI:** 10.3390/plants13091277

**Published:** 2024-05-06

**Authors:** Chenwei Zhang, Dandan Wang, Yan Li, Zifan Wang, Zhiming Wu, Qingyin Zhang, Hongwei Jia, Xiaoxu Dong, Lianfen Qi, Jianhua Shi, Zhonglin Shang

**Affiliations:** 1Shijiazhuang Academy of Agricultural and Forestry Sciences, Shijiazhuang 050041, China; cwzhang4192@foxmail.com (C.Z.); yuwenhanzhu@126.com (D.W.); yoginlily@126.com (Y.L.); wzfwan666@163.com (Z.W.); xli760103@163.com (Q.Z.); jiahongwei_dxx@163.com (H.J.); nkyqlf@163.com (L.Q.); 2Hebei Collaboration Innovation Center for Cell Signaling and Environmental Adaptation, Hebei Research Center of the Basic Discipline of Cell Biology, College of Life Sciences, Hebei Normal University, Shijiazhuang 050024, China; 3Modern Agricultural Science and Technology Laboratory, Shijiazhuang University, Shijiazhuang 050035, China; 4Institute of Cash Crops, Hebei Academy of Agricultural and Forestry Sciences, Shijiazhuang 050031, China; hbnkywzm@163.com; 5College of Agricultural and Forestry Technology, Hebei North University, Zhangjiakou 075000, China; xxwz_jswh@163.com

**Keywords:** tomato (*Solanum lycopersicum*), *tomato yellow leaf curl virus* (TYLCV), gibberellin, reactive oxygen species, functional gene expression

## Abstract

*Tomato yellow leaf curl virus* (TYLCV) is a prominent viral pathogen that adversely affects tomato plants. Effective strategies for mitigating the impact of TYLCV include isolating tomato plants from the whitefly, which is the vector of the virus, and utilizing transgenic lines that are resistant to the virus. In our preliminary investigations, we observed that the use of growth retardants increased the rate of TYLCV infection and intensified the damage to the tomato plants, suggesting a potential involvement of gibberellic acid (GA) in the conferring of resistance to TYLCV. In this study, we employed an infectious clone of TYLCV to inoculate tomato plants, which resulted in leaf curling and growth inhibition. Remarkably, this inoculation also led to the accumulation of GA_3_ and several other phytohormones. Subsequent treatment with GA_3_ effectively alleviated the TYLCV-induced leaf curling and growth inhibition, reduced TYLCV abundance in the leaves, enhanced the activity of antioxidant enzymes, and lowered the reactive oxygen species (ROS) levels in the leaves. Conversely, the treatment with PP333 exacerbated TYLCV-induced leaf curling and growth suppression, increased TYLCV abundance, decreased antioxidant enzyme activity, and elevated ROS levels in the leaves. The analysis of the gene expression profiles revealed that GA_3_ up-regulated the genes associated with disease resistance, such as WRKYs, NACs, MYBs, Cyt P450s, and ERFs, while it down-regulated the DELLA protein, a key agent in GA signaling. In contrast, PP333 induced gene expression changes that were the opposite of those caused by the GA_3_ treatment. These findings suggest that GA plays an essential role in the tomato’s defense response against TYLCV and acts as a positive regulator of ROS scavenging and the expression of resistance-related genes.

## 1. Introduction

As they grow, plants confront an array of biotic stresses, including infections caused by viruses, bacteria, and fungi. Among these adversaries, viral diseases pose a significant threat to plant growth and reproduction [1]. Viruses infect plants and trigger the synthesis of pathogenic proteins, which disrupt host cell metabolism, resulting in reduced photosynthesis, increased respiration, and the accumulation of toxic compounds [2]. The cultivation of “virus-free” plants or the use of virus-resistant plant varieties can effectively mitigate the damage caused by viral attacks. Whiteflies, which are known vectors of multiple plant viruses, can be controlled by using chemical (insecticides), physical (quarantine nets), or biological (natural enemies) methods to reduce viral infection rates [3,4]. However, some control measures are less effective, and chemical treatments may lead to the presence of pesticide residues in agricultural products and the environment. To elucidate the mechanisms underlying resistance and to develop measures to reduce the incidence of viral diseases, it is essential to understand the physiological responses of plants to viral infections.

In response to viral infections, plant cells activate a monitoring system within their cell walls. This system triggers the accumulation of enzymes that scavenge reactive oxygen species and various reductive metabolites [5], mitigating damage in the early stages of infection [6]. Virus-infected plant cells employ multiple recognition receptors and signaling agents, which are located in the plasma membrane or cytoplasm, to initiate defensive responses, including the hypersensitive response (HR) or programmed cell death [7]. Viral infection sometimes leads to the alteration of phytohormones in plant cells. For instance, *Mal de Río Cuarto virus* (MRCV) induces the accumulation of jasmonic acid (JA), brassinolide (BR), abscisic acid (ABA), and indole acetic acid (IAA) in the leaf cells of maize and wheat; the expression of the proteins related to phytohormone metabolism and transport is also significantly increased [8]. The *Rice black stripe dwarf virus* (RBSDV) seriously disturbs phytohormone metabolism in rice; the ABA and cytokinin (CTK) contents increase, while the IAA, gibberellin (GA), JA, and salicylic acid (SA) contents decrease [9]. *Tomato chlorotic virus* (ToCV) infection leads to the mobilization of the SA and JA metabolic pathway components [10]. These results suggest that phytohormone metabolism may be a target of viral attacks and that certain phytohormones may play a role in activating the plant’s resistance mechanisms.

Phytohormones are intricately involved in the plant’s defensive responses to viral attacks, and increasing the levels of specific phytohormones may enhance resistance to viral diseases. For instance, the application of GA has been shown to improve antioxidant capacity and enhance resistance to multiple stresses [11,12,13]. In the case of *Plum pox virus* infection, the endogenous GA_3_ level in peaches increases, and increasing GA_3_ levels through grafting can effectively reduce viral replication [14]. Potato plants possess an sRNA-GA regulatory circuit that plays a vital role in virus defense, with small RNAs mediating crosstalk between GA signaling and SA signaling. The disruption of this circuit by *Potato virus Y* weakens the plant’s hypersensitive response and exacerbates viral disease symptoms [15]. These findings suggest that GA may act as a positive regulator in the resistance to viral attacks, but its precise role and mechanisms require further investigation.

TYLCV, a single-stranded DNA virus belonging to the Geminiviruses family, is a prominent pathogen that is responsible for tomato virus diseases. The whitefly *Bemisia tabaci* Mediterranean (MED) is mainly responsible for the rapid spread and mixed infection of TYLCV [3]. The TYLCV genome comprises six open reading frames, which encode six effector proteins [16]. Among these proteins, V1 and V2 are encoded by the positive-sense strand, while C1, C2, C3, and C4 are encoded by the antisense strand [17]. The capsid proteins are instrumental in viral replication and movement and in counteracting the plant’s defense responses. V2 has been identified as an RNA silencing suppressor; it is capable of binding double-stranded siRNA and destabilizing the proteins involved in the antiviral RNA silencing pathway. It also disrupts the function of several transcription factors in tomato cells [18,19,20]. Consequently, TYLCV severely impedes the growth and development of tomato plants, resulting in leaf yellowing, wrinkling, and reduced fruit-setting rates [21]. Novel strategies are urgently needed to reduce viral infections or to enhance the tomato plant’s ability to resist viral attacks.

In our preliminary research, we observed that tomato seedlings treated with PP333, a growth retardant, exhibited heightened sensitivity to viral attacks compared to untreated seedlings. Additionally, the abundance of TYLCV was significantly higher in the treated seedlings. These observations suggest that GA may be involved in tomato resistance to TYLCV. This study aimed to investigate the role of GA in tomato’s resistance to viral diseases and its possible underlying mechanisms.

## 2. Results

### 2.1. Agrobacterium-Mediated Infection Introduces TYLCV into Tomato Plants

Following the injection of *Agrobacterium tumefaciens*, which contained an infectious clone of TYLCV, into the tomato plants, a normal sharp peak was noticed in the melting curve, while several disordered peaks appeared in the plants into which an empty vector was injected, indicating that the infectious clone successfully introduced TYLCV into the tomato plants (Figure 1A) (a standard curve was recorded to show the dynamics of fluorescent signals as the temperature increased. The curve indicated a sudden drop in fluorescent signal at the melting temperature. If the PCR process yielded the desired product, a single-peak curve would be observed). TYLCV replication remained relatively stable from day 3~6 post-inoculation but then increased rapidly from the sixth day, reaching 3.28 ± 0.52 times and 3.76 ± 0.53 times the initial levels on days 9 and 12; these values were significantly higher than the initial value (*p* < 0.05) (Figure 1B). TYLCV was detected in apical leaves 20 days after inoculation, indicating the systemic movement of the virus in the plants (Figure 1C). New leaves were more sensitive to viral attack than mature leaves; hence, virus abundance and the physiological parameters in the apical leaves were investigated in the following experiment. 

### 2.2. TYLCV Infection Induces Phytohormones Accumulation

The concentration of phytohormones in the apical leaves was measured 20 days after inoculation. The IAA and ABA concentrations showed no significant changes at 25, 30, and 35 days post-inoculation. However, at 35 days post-inoculation, the SA content was 4.6 ± 0.18 times higher than that in the control (*p* < 0.05), and the JA content was 1.69 ± 0.17 times higher (*p* < 0.05). The trans-zeatin-riboside (TZR) content increased to a level that was 2.85 ± 0.46 times higher than that in the control at 25 days post-inoculation and then gradually decreased. The most significant change was observed in the GA_3_ content, which had increased by 69.48 ± 0.48 times at 30 days post-inoculation (*p* < 0.05) (Figure 2).

### 2.3. GA_3_ Level Is Positively Correlated with Tomato’s Resistance to TYLCV

To verify the role of GA_3_ in inducing virus resistance, the effects of added GA_3_ or PP333 on TYLCV abundance in apical leaves were investigated. Firstly, the tomato plants were inoculated with TYLCV using *Agrobacterium tumefaciens*. Twenty days after inoculation, GA_3_ (75 mg/L) or PP333 (0.25 mg/L) was sprayed on the systemic leaves; then, 5, 10, 15, 20, and 25 days after the spraying, the apical leaves were collected for TYLCV abundance detection. In the control plants sprayed with water, the virus abundance increased from days 5 to 15 and then decreased. After GA_3_ application, the virus abundance increased in a similar manner to that of the control from days 5 to 10 and then significantly decreased, reaching a very low level (less than 1/200 of the original level) after 5 days, with no marked rebound. In contrast, the PP333 treatment led to a significant increase in virus abundance from days 5 to 15, with an increment that was 41 ± 20% higher than that of the control, followed by a decrease to a level close to that of the control (Figure 3A). The symptoms in the control plants, such as leaf curling and yellowing, were significantly rescued by GA_3_ and enhanced by PP333 (Figure 3B).

### 2.4. GA_3_ Enhances Antioxidant Activity of Tomato Leaf Cells

To explore the role of GA_3_ in the regulation of ROS metabolism, ROS accumulation in the apical leaves was assessed using DAB staining. After TYLCV infection, the ROS concentration significantly increased. The addition of GA_3_ effectively reduced ROS accumulation, while PP333 enhanced ROS accumulation in the leaves (Figure 4A). Additionally, the activity of several antioxidant enzymes, including POD, CAT, and SOD, was detected in the apical leaves. The POD activity increased by about 35% between 5 and 15 days after GA_3_ application and decreased by 38% after PP333 treatment. The CAT activity changed in a similar manner and to a similar degree after GA_3_ or PP333 treatment. The SOD activity increased by 20% on the 10th day after GA_3_ treatment and decreased by 14% after PP333 treatment (Figure 4B). 

### 2.5. GA_3_ Alters Expression of Functional Genes

To further elucidate the mechanism of GA_3_ in the promotion of virus resistance, the expression pattern of functional genes in TYLCV-infected plants after GA_3_ or PP333 treatment was investigated using high-throughput RNA sequencing. The preliminary results revealed 606 up-regulated and 492 down-regulated differentially expressed genes (DEGs, |log_2_(FoldChange)| ≥ 3) after GA_3_ treatment and 607 up-regulated and 346 down-regulated DEGs after PP333 treatment (listed in Table 1 and Table 2) (Figure 5).

GO analysis showed that GA_3_-affected, up-regulated DEGs were involved in biological processes, cellular components, and molecular functions and were enriched in functions related to response to stimuli, metabolic processes, cellular processes, catalytic activity, etc. Down-regulated genes were enriched in functions related to catalytic activity, cellular processes, metabolic processes and biological regulation, response to stimuli, etc. (Figure 6A). Up-regulated DEGs induced by PP333 treatment were involved in biological processes, cellular components, and molecular function; they were enriched in functions related to response to stimuli, biological regulation, positive regulation of biological processes, and cellular component organization or biogenesis, transporter activity, and catalytic activity, among others. Down-regulated DEGs were enriched in functions related to response to stimuli, multi-organism processes, biological regulation, metabolic processes, catalytic activity, etc. (Figure 6B).

KEGG analysis showed that up-regulated DEGs induced by GA_3_ were enriched in the pathways related to p53 signaling; the degradation of valine, leucine, and isoleucine; protein processing in the endoplasmic reticulum; the cell cycle; DNA replication; Fanconi anemia; starch and sucrose metabolism; phenylpropanoid and cyanoamino acid metabolism; the MAPK signaling pathway; etc. Down-regulated DEGs were enriched in the pathways related to spliceosome, peroxisome, oxidative phosphorylation, glutathione metabolism, and drug metabolism (Figure 7A). In the PP333-treated plants, up-regulated DEGs were enriched in the pathways related to p53 signaling, ubiquitin-mediated proteolysis, pyrimidine metabolism, protein processing in the endoplasmic reticulum, the cell cycle, Fanconi anemia, and others. Down-regulated DEGs were enriched in the pathways related to starch and sucrose metabolism, phenylpropanoid biosynthesis, cyanoamino acid metabolism, peroxisome, ABC transporters, the gap junction, and others (Figure 7B).

Responding to GA_3_, the expressions of dozens of transcription factors (TFs) and disease resistance-related genes altered significantly. Among the up-regulated DEGs, WRKY TFs, ethylene-responsive TFs, bHLH TFs, MYB TFs, and NAC domain-containing proteins were revealed to be involved in plant defensive or hypersensitive responses, as well as the modulation of ROS metabolism (Table 1). The down-regulated DEGs included Cytochrome P450, MYB TFs, WRKY TFs, serine/threonine protein kinases, and heat shock proteins. The expression of the DELLA protein was significantly down-regulated, confirming the involvement of GA signaling in virus resistance (Table 1). Responding to PP333, genes including some Cytochrome P450 members, serine/threonine protein kinases, MYB TFs, WRKY TFs, B3 domain-containing TFs, and bHLH TFs were significantly down-regulated. Meanwhile, some Cytochrome P450 members, MYB TFs, NAC domain-containing proteins, heat shock proteins, serine/threonine protein kinases, ERF TFs, and ABC transporters were markedly up-regulated (Table 2).

It is noteworthy that the expressions of some of the genes were stimulated by GA_3_ while they were suppressed by PP333. The expressions of these genes were further detected using real-time qPCR. The results confirmed that Cytochrome P450 94C1, MYB117, MYB20, a MADS-box TF (MADS-MC), Carrot ABA-induced in somatic embryos 3-like, Mitogen-activated protein kinase kinase kinase 18 (MAPKKK18)-like, Pathogenesis-related protein PR-5 precursor, and Wound-induced proteinase inhibitor 1 precursor were significantly up-regulated by GA_3_, while they were down-regulated by PP333 in TYLCV-infected tomato plants (Figure 8). The functions of these genes, e.g., MYB117, MYB20, pathogenesis-related protein PR-5 precursor, and wound-induced proteinase inhibitor 1 precursor, are closely related to plant disease resistance (Table 3).

## 3. Discussion

As a primary pathogen, TYLCV significantly disrupts tomato growth and reproduction. Studying how tomato plants respond to viral infection can offer valuable insights for the development of strategies to reduce the reproductive and quality losses caused by viral diseases. In field tomato production, the application of growth retardants is essential to control plant overgrowth, which can impact flowering and fruiting. However, we observed that the use of growth retardants can increase the incidence and severity of viral diseases in tomato plants. This phenomenon is possibly due to the inhibition of GA synthesis. This suggests that GA might play a crucial and positive role in the triggering of virus resistance in tomato plants. To test this hypothesis, the role of GA in the response to TYLCV was investigated.

Viral infections are known to induce changes in phytohormone levels in various crops. For instance, infection with the Rice dwarf virus led to a significant decrease in GA content [22]. RBSDV infection resulted in elevated levels of ABA and CTKs and a decrease in the levels of IAA, GA, JA, and SA in rice [9]. Infection with the *Ageratum leaf curl Sichuan virus* (ALCScV) caused a decrease in GA content in tobacco leaves, accompanied by dwarf symptoms and abnormal flower development [23]. Viral genome-directed protein synthesis can disrupt phytohormone signal transduction, which is responsible for the eliciting of resistance to pathogens. For instance, the C4 protein encoded by the ALCScV genome interacts with the DELLA protein, blocking GA signaling in tobacco plants [23]. The P7-2 protein encoded by RBSDV interacts with the GA receptor (GID2) in rice and maize, hindering signal transduction from GID2 to downstream targets [24]. These findings suggest that viruses may disrupt GA metabolism and signal transduction early in the infection process, weakening the plant cells’ defensive capabilities and creating conditions conducive to viral replication. In this study, TYLCV was found to affect the levels of several phytohormones, with GA accumulating significantly in the virus-infected plants. This suggests that GA may play a central role in the triggering of the virus-resistant response. Notably, this study provides novel insights into the changes in GA metabolism in response to TYLCV infection; such an increase in endogenous GA in response to TYLCV infection has not previously been reported. 

In this research, the addition of GA or PP333 to tomato plants further confirmed that GA plays a positive role in the enhancement of antiviral activity. Previous studies have reported that added GA_3_ alleviated the dwarfing symptoms caused by RBSDV [9]; the data in this study showed that GA may evoke a resistance response to TYLCV, further illustrating the role of this phytohormone in the biotic stress response. The GA signaling pathway is closely related to SA-induced resistance responses [25,26]. Overexpression of the MAPK4 gene in tobacco resulted in a significantly lower endogenous GA content than in the wild-type plants. These overexpression lines were more susceptible to infections with *Ralstonia solanacearum* and *Rhizoctonia solani*, indicating a positive correlation between the endogenous GA content and the resistance to these pathogens [27,28]. A mixture comprising bioprotein, bricin, IAA, and GA was reported to enhance tomato plants’ resistance to viral diseases, reducing virus-induced curling and chlorosis. However, the role of GA in the response to viral infection was not investigated [4]. In recent years, GA was primarily considered to be a growth regulator, with roles in growth, development, and the repair of damage caused by viruses through the promotion of cell proliferation and enlargement. Its role in defending against viral attacks has rarely been explored. The results in this study, which show that TYLCV replication was suppressed by GA_3_ and enhanced by PP333, along with the observation that GA_3_ alleviated and PP333 exacerbated the virus-induced disease symptoms, confirm that GA plays an essential and positive role in the triggering of antiviral responses in tomatoes.

In response to biotic and abiotic stresses, plants generate reactive oxygen species (ROS) within their cells to induce defensive responses. However, excessive ROS accumulation can harm cell structures and physiological functions. Antioxidant enzymes such as CAT, POD, and SOD play a crucial role in the scavenging of ROS [29,30,31]. Phytohormones are involved in the enhancement of resistance to biotic and abiotic stress through the stimulation of the activity of antioxidant enzymes. For instance, TYLCV infection led to an increased synthesis of POD, CAT, and SOD [32], and the addition of SA promoted the activity of POD and SOD, resulting in reduced TYLCV replication [33]. JA stimulated POD and CAT activity, improving drought stress tolerance in Brassica [34] and *Dioscorea zingiberensis* [35]. SA stimulated POD and CAT activity in mung beans and enhanced drought resistance [36]. GA_3_ was found to promote the activities of POD and CAT, enhancing salt tolerance in sorghum [37]. Additionally, the addition of GA_3_ stimulated POD, SOD, and CAT activity, which may be involved in rice’s resistance to *Nilaparvata lugens* [38]. Vanillin B1, a compound that increased the activity of POD, SOD, and CAT, enhanced resistance to the *Cucumber mosaic virus* (CMV) [39]. The results in this study indicate that TYLCV infection may lead to ROS accumulation, potentially causing damage to tomato leaf cells. The addition of GA_3_ stimulated ROS-scavenging enzymes, leading to a reduction in ROS levels, while PP333 inhibited these enzymes, leading to increased ROS levels. These results are consistent with the protective effect of GA_3_ against TYLCV-induced damage. The role of GA in the regulation of ROS scavenging in the resistance response to TYLCV infection has not been reported, and this study provides novel insights into the mechanism behind tomato plants’ resistance to viral attacks.

To further understand the physiological mechanisms of GA-enhanced virus resistance, the gene expression patterns affected by GA_3_ or PP333 were investigated. Previous reports have shown that, in TYLCV-infected plants, the expression of certain P450 family members (e.g., BTA009037.1 and BTA015105.1) was significantly up-regulated, while other P450 family members (e.g., BTA025848.1 and BTA015105.1) were markedly down-regulated. In addition, a gene encoding SOD(e.g., BTA025848.1) was down-regulated following TYLCV infection, which is consistent with increased ROS accumulation in plants [3]. Furthermore, the expression levels of various proteins such as heat shock protein (e.g., BTA025691.1), Cathepsin B (e.g., BTA015120.1), transmembrane protein (e.g., BTA001187.1), ATP synthase gamma chain (e.g., BTA007709.1), etc. have shown significant changes. However, it is important to note that despite the significant changes, none of these genes mentioned above coincide with the experimental results presented in this paper. Changes in gene expression induced by GA have been reported in previous studies. For instance, in *Abrus cantoniensis* seeds, the expression of genes such as CYP78A5, MYB4, LEA, CHS, and STH-2 was correlated with the addition of GA_3_ [28]. In the leaves of *Boehmaria nivea* L. treated with GA_3_ or ethylene, genes such as PAL, 4CL, CCR, CAD, C3H, and CYP98A were up-regulated, while others such as 4CL, HCT, and CYP73A were down-regulated [40]. We hypothesized that spraying GA_3_ would reduce virus accumulation by changing the expression of disease-resistant genes. This change in gene expression would enhance the plant’s ability to resist virus invasion. As a result, virus accumulation levels would decrease when the virus is present while also having a preventive effect when there is no virus. However, the effects of GA_3_ on gene expression were not investigated in plants in which the application of GA_3_ alleviated dwarfing [40]. In this study, the expression of the P450, NAC, WRKY, DELLA, and ERF genes was up-regulated by GA3. It has been reported that some of these, such as the WRKY and NAC family members, participate in SA-mediated defense responses [41]. The increased expression of NAC genes led to increased activity of SOD, POD, and CAT, and NACs were involved in GA signal transduction through interaction with DELLA proteins [26,42]. WRKY members, which can be up-regulated by both SA and GA, were involved in rice’s resistance to *Xanthomonas oryzae* [43]. It has been reported that some P450 family members are involved in the synthesis of antiviral and antibacterial substances and cell protectants, such as CYP94C1 in Arabidopsis, which participates in JA-induced resistance to *Botrytis cinerea* [44,45]. Some ERFs are involved in JA-induced defense responses [46]. The crosstalk between JA and GA signaling is well documented in plant stress tolerance and resistance [25]. In this experiment, the DELLA gene was down-regulated by GA_3_ treatment, confirming the involvement of GA signaling in tomato plants’ virus resistance. NACs, WRKYs, Cytochrome P450 members, and ERFs, which showed increased expression in response to GA, likely play a positive role in GA-induced virus resistance. The genes whose expression was up-regulated by GA_3_ and down-regulated by PP333 included Cytochrome P450 94C1, MYB117, MYB20, and PR-5 precursor, among others. These proteins have been implicated in SA- or JA-induced defense responses, allergic reactions, and secondary metabolite production [44,47,48,49,50,51,52]. The roles of these proteins in enhancing resistance to TYLCV were confirmed in this study, and these results were consistent with those of previous studies. Furthermore, the results provided new insights into the regulation of the expression of these genes.

In addition to GA, TYLCV infection increased the levels of SA, JA, and TZR in this experiment. It is well known that the SA and JA antivirus pathways may be activated when viruses invade the plants [53,54]. The TZR content could be influenced by abiotic stress, e.g., drought and salinity. The role of TZR in plant cell division and expansion has been well illustrated; to date, however, the effect of viral infection on the TZR level in crops has rarely been investigated [55,56,57]. In this study, TYLCV infection led to an increased TZR level, suggesting that TZR might be correlated with plant stress resistance, although the role and functional mechanism of TZR in the plant’s antiviral response needs to be further investigated. 

Based on the abovementioned results, it can be concluded that GA plays a critical role in tomato plants’ resistance to TYLCV; therefore, the use of growth retardants may significantly increase the risk of TYLCV infection. Strict control of the dosage of growth retardants is essential. As the application of growth retardants is a necessary measure in tomato cultivation, it is vital to explore new approaches that may be used to induce virus resistance in seedlings following growth retardant treatment. Investigating the effects of various plant growth substances on the virus resistance of tomato plants may provide valuable insights for the development of effective measures.

## 4. Materials and Methods

### 4.1. Plant Materials

The tomato plants (*Solanum lycopersicum* L., cv. Nongbofen 18109) were grown in a greenhouse under controlled conditions at 26 ± 2 °C, 80 ± 5% relative humidity, and a photoperiod of 15/9 h light/dark. Isolation nets were used to protect the plants from whiteflies, and no common tomato viruses, including TYLCV, ToMV (tomato mosaic virus), and ToCV (tomato chlorosis virus), were detected in the seeds and seedlings tested.

### 4.2. TYLCV Infection

TYLCV was introduced into tomato plants using a previously reported infectious clone [58] (GenBank: KF612971.1). *Agrobacterium tumefaciens* EHA105, carrying the TYLCV infectious clone, was cultured in LB medium containing 100 mg·L^−1^ rifampicin (Rif) and 50 mg·L^−1^ kanamycin sulfate (Kan) for 48 h. The bacteria suspension was centrifuged at 4 °C at 5000 rpm for 5 min. The sediment was re-suspended and diluted with an inoculum consisting of 10 mmol·L^−1^ MES, 10 mmol·L^−1^ MgCl_2_, and 150 μmol·L^−1^ acetylsyringone. Following the method of Huang et al. [32], the diluted bacteria suspension (OD600 value 0.6~0.8) was prepared and injected into tomato leaves with a syringe without a needle. In each treatment, at least 10 tomato plants were inoculated, and the data from 3 replicates were collected and calculated. 

### 4.3. Measurement of Endogenous Phytohormone Level 

The concentrations of GA, TZR, IAA, ABA, SA, and JA in the tomato leaves were determined by Nanjing Ruiyuan Biotechnology Co., Ltd. (Nanjing, China). Following the reported method [59], the phytohormones were extracted with acetonitrile and were then detected using high-performance liquid chromatography (AGLIENT1260, Agilent Technologies Inc., Santa Clara, CA, USA) and tandem mass spectrometry (AB4000, SCIEX Inc., Framingham, MA, USA). An amount of 1.5 g of tomato leaf tissue was ground in liquid nitrogen and mixed with 15 mL of acetonitrile; after being stored at 4 °C for 12 h, the extract was centrifuged at 4 °C at 12,000 rpm for 5 min. The collec ted supernatant was mixed with an appropriate amount of C18 and GCB (graphitized carbon black) and centrifuged at 4 °C at 12,000 rpm for 5 min. The supernatant was collected and blow-dried with nitrogen gas and redissolved with 400 μL of methanol. The solution was filtered with a 0.22 μm filter (Millipore SLGNX13NL, Merck KGaA, Darmstadt, Germany) and stored at −20 °C. Poroshell 120 SB-C18 reversed-phase chromatography was adopted for the analysis. The data from 3 replicates were collected and calculated.

### 4.4. DNA, RNA Extraction, and qPCR

The tomato DNA was extracted to assess the replication level of TYLCV following the reported method [60]. The Ezup column superplant genome DNA extraction kit was used for DNA extraction. Total RNA was extracted to verify the variation trend of the endogenous gene expression in the tomato leaves using the RNeasy Plant Mini Kit (Qiagen, Dusseldorf, Germany). cDNA synthesis, primed with random primers, was carried out at 42 °C using 500 ng of total RNA. The DNA and RNA content was quantified using a spectrometer (Nano-Drop 2000, Thermo Fisher Scientific, Waltham, MA, USA), and the quality was assessed using agarose gel electrophoresis. The DNA and cDNA samples were diluted (1:5) with sterile diethylpyrocarbonate-treated water. Quantitative PCR(qPCR) to measure the virus levels in the plant tissues was conducted using the SYBR PrimeScript^TM^ real-time PCR Kit (Noweizan, Nanjing, China), following the reported method [61]. The detection was performed with a CFX96 qPCR system (Bio-Rad, Hercules, CA, USA). The accumulation of TYLCV was detected using the primers TYLCV-Q-F (TAATCATTTCCACGCCCGTCTC) and TYLCV-Q-R (CAGTATGCTT AATATCATCCCGTTGCTC). Other gene-specific primers, which were used in the verification analysis of the up-regulation or down-regulation of the genes in the results from the RNA-seq, were designed using DNAMAN 8.0 software. The primers are listed in Appendix A. After the amplification, a melting curve analysis was performed to distinguish the specific virus sequence from the other products.

### 4.5. Determination of ROS in Tomato Leaves 

The levels of reactive oxygen species (ROS) in the tomato leaf cells were determined using 0.1% DAB (diaminobenzidine) staining. The tomato leaves were soaked in a 0.1% DAB solution (dissolved in 50 mmol·L^−1^ Tris-HCl buffer, pH adjusted to 3.8) at 25 °C for 6–8 h. Subsequently, the leaves were washed with water and boiled in 95% ethanol for 15 min, followed by soaking in fresh 95% ethanol for 12–18 h until the chlorophyll was completely faded. The stained leaves were then photographed using a stereomicroscope (SZX16, Olympus Corporation, Tokyo, Japan) equipped with a digital camera.

### 4.6. Determination of Antioxidant Enzyme Activity

The activity of the peroxidase (POD), superoxide dismutase (SOD), and catalase (CAT) in the tomato leaves was determined by Nanjing Ruiyuan Biotechnology Co., LTD. (Nanjing, China), using the corresponding enzyme activity detection kits (Solarbio, Beijing, China) and following the reported methods [62,63,64]. 

An amount of 0.1 g of tomato leaf tissue was ground in 1 mL of phosphate buffer solution (PBS) (100 mmol·L^−1^, pH 6.0) in an ice bath. After being centrifuged at 4 °C at 8000 rpm for 10 min, the supernatant was collected as the sample solution. 

For the POD activity measurement, the working solution was prepared by mixing PBS, 2-methoxyphenol, and 30% H_2_O_2_ in a ratio of 2.6 (mL):1.5 (μL):1 (μL). A 10 μL sample solution was mixed with a 190 μL working solution, and the light absorption values of A1 at 470 nm for 1 min and A2 after 2 min were detected using a DNM9606 microplate reader (Perlong, Beijing, China).

For the CAT activity measurement, 20 μL of sample solution was mixed with 100 μL of Tris buffer (100 mmol·L^−1^, pH 7.8) and incubated at 25 °C for 10 min. After adding 100 μL (NH_4_)_2_MoO_4_ (100 mmol·L^−1^), the solution was incubated for 10 min, and the light absorption value at 405 nm was detected. 

For the SOD activity measurement, 50 μL of sample solution was mixed with 1.5 mL of Tris buffer (100 mmol·L^−1^, pH 7.8), 1.5 mL of methionine solution (130 mmol·L^−1^), 0.3 mL of EDTA∙Na_2_ solution (100 μmol·L^−1^), 0.3 mL of riboflavin solution (20 μmol·L^−1^), and 0.3 mL of nitrogen blue tetrazole (750 μmol·L^−1^). The mixture was incubated at 25 °C for 10 min. The light absorption value at 506 nm was detected.

### 4.7. RNA Sequencing Analysis

Total RNA from the tomato leaves was extracted using the Qubit RNA Extractor (Trizol) RNA extraction kit (Sangon Biotech, Shanghai, China). The RNA concentration and integrity were assessed using Qubit 2.0 and agarose gel electrophoresis, respectively. mRNA separation, cDNA library construction, and transcriptome sequencing were performed by Sangon Biotech Co., Ltd. (Shanghai, China) using the Illumina HiSeq^TM^ sequencing platform. The tomato SL4.0 genome was used as the reference for sequence comparison. The potential functions of the differentially expressed genes were analyzed using gene ontology (GO) enrichment and Kyoto Encyclopedia of Genes and Genomes (KEGG) enrichment analysis. 

### 4.8. Data Analysis

Curve plotting was performed using Microsoft Excel 2016, and data analysis to calculate the significant differences was carried out using SPSS software (SPSS Inc., Chicago, IL, USA).

## Figures and Tables

**Figure 1 plants-13-01277-f001:**
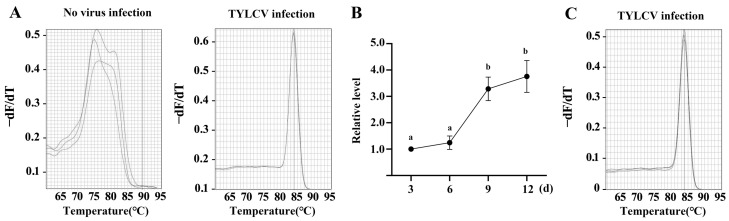
TYLCV infection and transport in tomato plants. (**A**) Analysis of the TYLCV melting curve. DNA was extracted from infected leaves 3 days after inoculation and analyzed using real-time PCR (the *y*-axis indicates the negative first derivative of the normalized fluorescence generated during PCR amplification). (**B**) Dynamics of virus abundance in infected leaves, with relative abundance of the TYLCV CP gene detected using qPCR. Different lowercase letters within the data are significantly different based on the Duncan multiple range test at *p* < 0.05. (**C**) Melting curve analysis of TYLCV in apical leaves 20 days after inoculation.

**Figure 2 plants-13-01277-f002:**
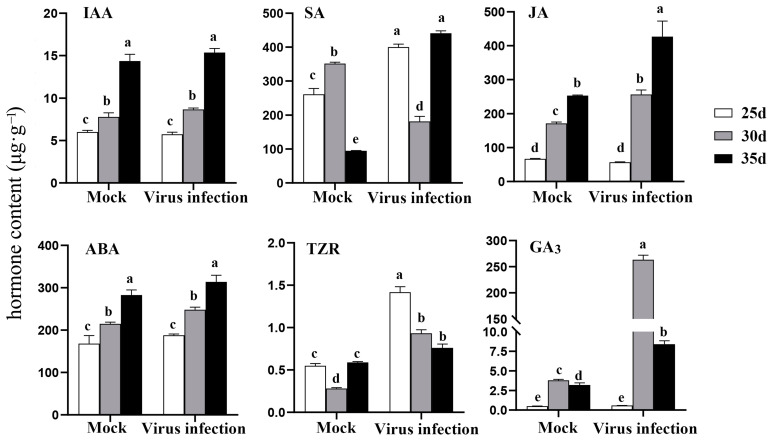
TYLCV infection affected endogenous phytohormone levels in tomato apical leaves. IAA, SA, JA, ABA, TZR, and GA_3_ contents were assessed in apical leaves at 25, 30, and 35 days after inoculation. The vertical axis represents hormone content (μg/g fresh weight). Different lowercase letters within any data are significantly different based on the Duncan multiple range test at *p* < 0.05.

**Figure 3 plants-13-01277-f003:**
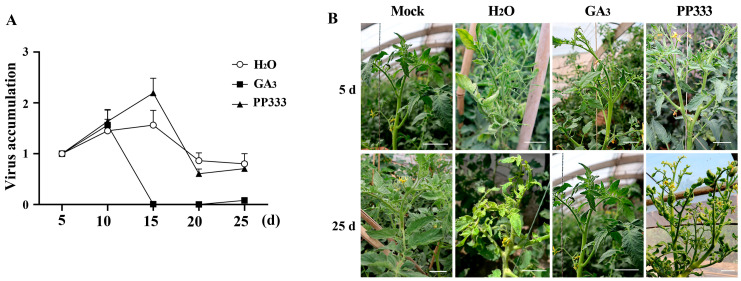
Effects of GA_3_ or PP333 on TYLCV abundance in tomato leaves. (**A**) Dynamics of virus abundance following GA_3_ or PP333 spraying. (**B**) Leaf symptoms at 5 and 25 days after different treatments. Scale bar = 5 cm.

**Figure 4 plants-13-01277-f004:**
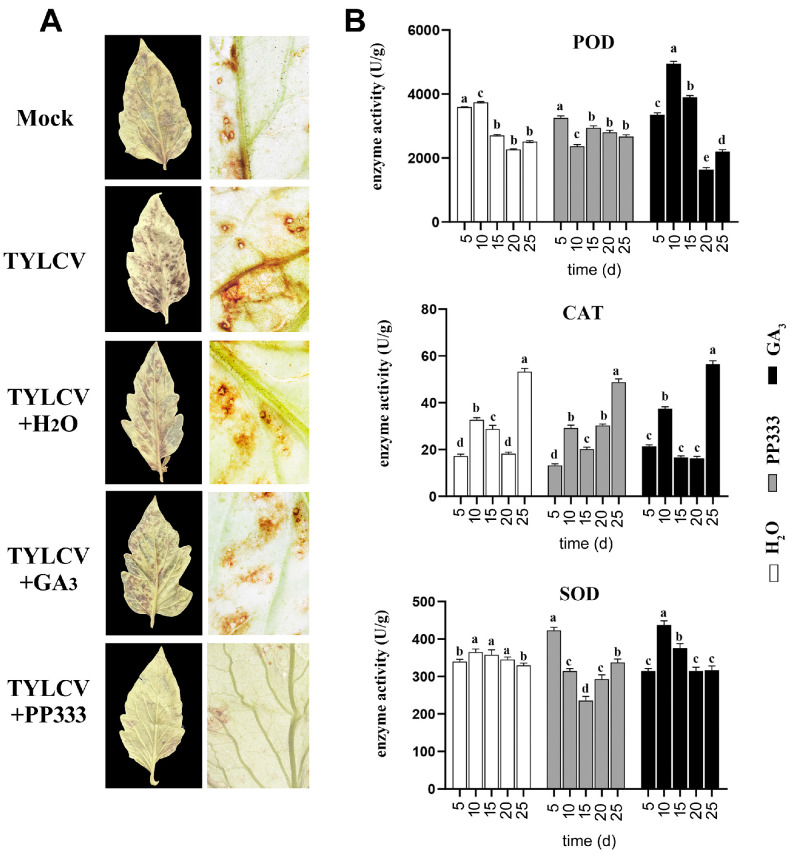
Influence of GA_3_ or PP333 on ROS accumulation and antioxidant enzyme activity in apical leaves. Tomato plants infected with TYLCV were treated with GA_3_ or PP333 20 days after infection. Fifteen days later, apical leaves were collected for DAB staining and antioxidant enzyme assessment. (**A**) Photographs of DAB-stained apical leaves, including whole leaf images (left column) and enlarged leaf sections (right column). (**B**) Activity levels of POD, CAT, and SOD in apical leaves. Different lowercase letters within any data are significantly different based on the Duncan multiple range test at *p* < 0.05.

**Figure 5 plants-13-01277-f005:**
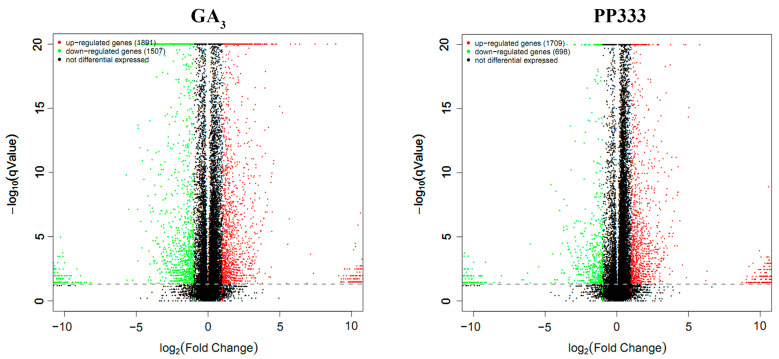
Differentially expressed genes in apical leaves of TYLCV-infected plants before and after GA_3_ or PP333 treatment. Differentially expressed genes (DEGs) before and after GA_3_ (**left**) or PP333 (**right**) treatment are shown. Tomato plants were infected with TYLCV and treated with GA_3_ or PP333 20 days later, and RNA was extracted from apical leaves for RNA sequencing 7 days post-treatment. In the figure, up-regulated genes are represented by red dots, down-regulated genes by green dots, and genes with no significant change by black dots. The *x*-axis indicates the fold change, and the *y*-axis represents −log10 (qValue).

**Figure 6 plants-13-01277-f006:**
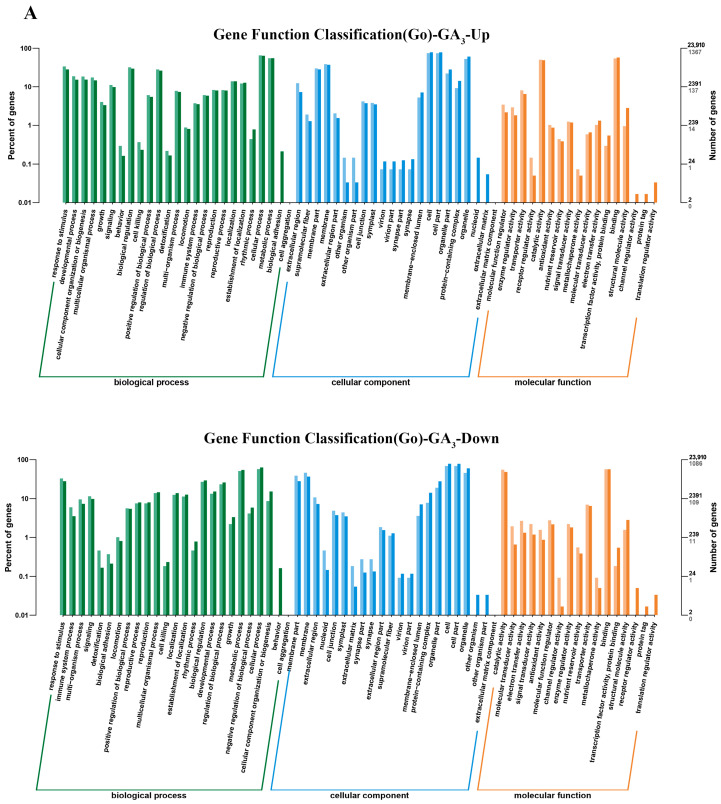
Gene ontology (GO) analysis of DEGs in GA_3_- or PP333-treated tomato leaves. (**A**,**B**) show GO enrichment analysis results of up-regulated and down-regulated DEGs in GA_3_- (**A**) or PP333-treated (**B**) plants, respectively.

**Figure 7 plants-13-01277-f007:**
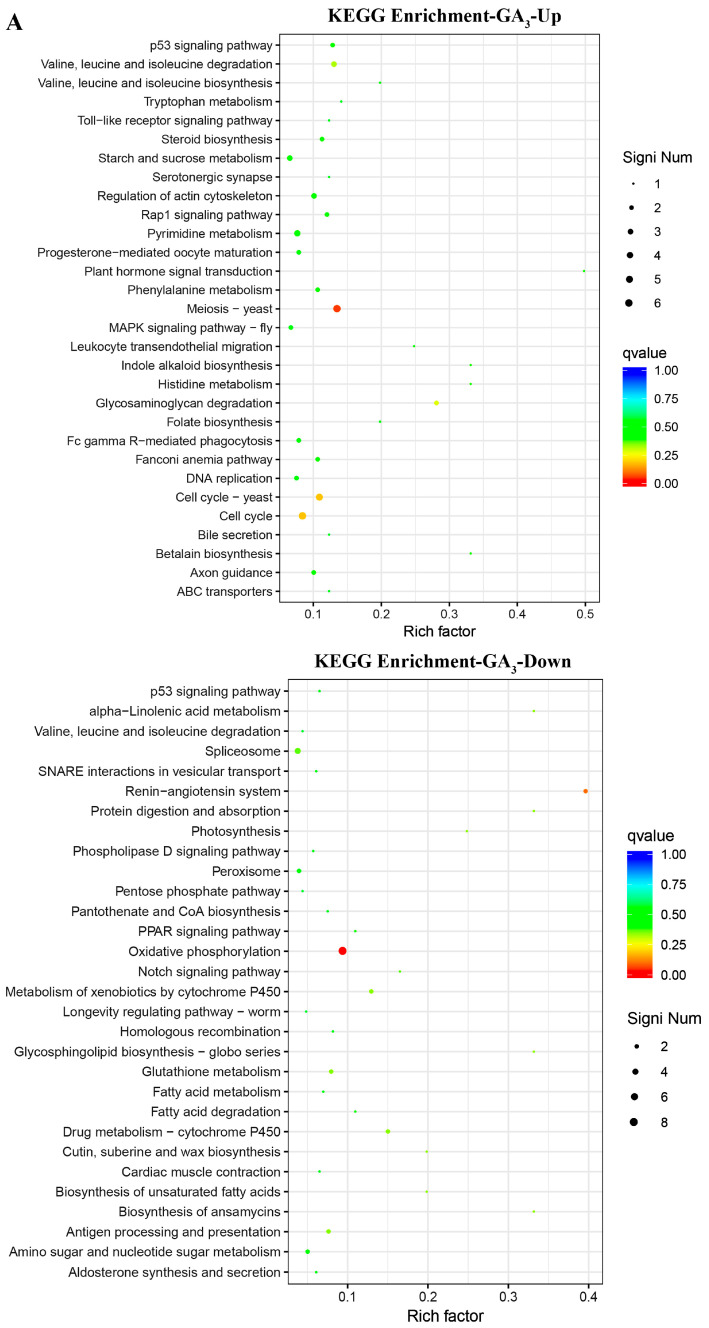
KEGG analysis of DEGs in GA_3_- or PP333-treated tomato leaves. KEGG enrichment analysis results of up-regulated and down-regulated DEGs in GA_3_- (**A**) or PP333-treated (**B**) plants are shown, respectively.

**Figure 8 plants-13-01277-f008:**
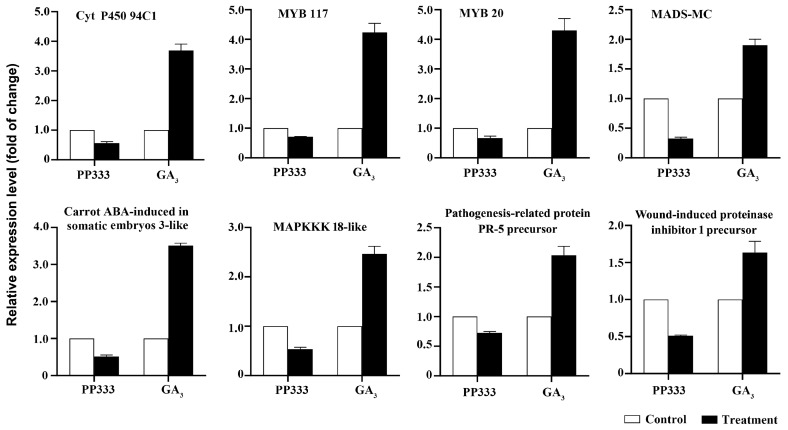
Expression of several genes which were up-regulated by GA_3_ and down-regulated by PP333. Real-time qPCR was used to detect gene expression in apical leaves of tomato plants, which were infected with TYLCV and then treated with GA_3_ or PP333.

**Table 1 plants-13-01277-t001:** GA_3_-regulated DEGs in tomato leaves.

Up-Regulated Gene Annotation Group	Number of Genes
Cytochrome P450	13
MYB transcription factors	10
B3 domain-containing transcription factors	2
NAC domain-containing proteins	3
Acyl-CoA synthetases	7
Ca^2+^-dependent phospholipid-binding proteins	8
Serine/threonine protein kinases	10
LOB domain-containing proteins	2
Ethylene-responsive transcription factors	11
Mitogen-activated protein kinases	4
MADS-box transcription factors	3
Heat shock proteins	3
Sugar transport proteins	2
Acylsugar acyltransferases	4
Other	524
**Down-Regulated Gene Annotation Group**	
Cytochrome P450	12
MYB transcription factors	9
WRKY transcription factors	3
B3 domain-containing transcription factors	4
F0F1-type ATP synthases	3
Calmodulin and related proteins	6
Ribosomal proteins	2
Serine/threonine protein kinases	15
Acyl-CoA synthetases	4
Pathogenesis-related proteins	3
bHLH-like transcription factors	4
Jasmonic acid-amido synthetase	1
Heat shock proteins	9
DELLA	1
Other	417

**Table 2 plants-13-01277-t002:** PP333-regulated DEGs in tomato leaves.

Up-Regulated Gene Annotation Group	Number of Genes
Cytochrome P450	9
MYB transcription factors	11
B3 domain-containing transcription factors	3
NAC domain-containing proteins	5
Acyl-CoA synthetases	9
bHLH transcription factors	5
Serine/threonine protein kinases	15
LOB domain-containing proteins	3
Ethylene-responsive transcription factors	8
Mitogen-activated protein kinases	5
MADS-box transcription factors	4
Heat shock proteins	3
ABC transporters	7
Other	520
**Down-Regulated Gene Annotation Group**	
Cytochrome P450	7
MYB transcription factors	4
WRKY transcription factors	2
B3 domain-containing transcription factors	4
bHLH transcription factor	1
Ribosomal proteins	2
Serine/threonine protein kinases	11
Acyl-CoA synthetases	5
Pathogenesis-related proteins	3
Ethylene response factors	2
Nuclear transcription factors	2
Heat shock proteins	9
Other	294

**Table 3 plants-13-01277-t003:** Function annotation of the genes that were up-regulated by GA_3_ and down-regulated by PP333.

Gene Identification	Gene Description	Function (Information Was Obtained from the National Center for Biotechnology Information (NCBI) Website.)
Solyc06g074420.1	cytochrome P450 94C1	Related to plant stress response, JA defense response pathway, and cell senescence
Solyc05g007870.3	MYB117	Related to plant disease resistance, plant reproduction, and growth
Solyc05g014290.4	MYB20	Related to plant disease resistance and plant cell wall synthesis
Solyc05g056620.2	MADS-box TF (MADS-MC)	Related to plant fruit ripening
Solyc09g014750.1	Carrot ABA-induced in somatic embryos 3-like	Related to plant fruit ripening
Solyc07g051890.1	Mitogen-activated protein kinase kinase kinase 18-like	Related to cell mitosis and plant signal transduction
Solyc08g080660.1	Pathogenesis-related protein PR-5 precursor	Associated with SA defense reaction, plant allergic reaction, and plant oxidation
Solyc09g084470.3	Wound-induced proteinase inhibitor 1 precursor	Related to plant stress response and plant wound

## Data Availability

Data available on request from the authors.

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
