# Peer review of "Gibberellin Positively Regulates Tomato Resistance to Tomato Yellow Leaf Curl Virus (TYLCV)"

_plants, 2024, doi:10.3390/plants13091277_

Round 1
Reviewer 1 Report
Comments and Suggestions for Authors
Dear authors,
The study focusing on the Gibberellic Acid (GA) resistance to TYLCV virus presents valuable insights for future research. However, I have some concerns and suggestions that need to be addressed before accepting the paper.
1. It would be beneficial for the study to include the GA expression pattern in susceptible varieties when exposed to TYLCV infection. This comparison would help determine if there are any differences between resistant and susceptible varieties regarding GA expression levels.
2. I recommend merging the steps for DNA extraction and RT-PCR, as well as RNA extraction and RT-PCR to streamline the process. Additionally, consider changing "RT-qPCR" to maintain consistency in terminology.
3. It would enhance the discussion to include previous studies conducted on different viruses using GA function in other plant species. This comparative analysis would provide additional context and support for the findings presented in the paper.
Kindly address these revisions in your manuscript before resubmitting.
Reviewer 2 Report
Comments and Suggestions for Authors
The manuscript "Gibberellin positively regulates tomato's resistance to Tomato Yellow Leaf Curl Virus" by Zhang et al, deals with the observation that adding gibberellin alters the tomato response to a monopartite begomovirus. This observation is relevant due to the use of these plant hormones in the horticulture industry. The authors artificially inoculated tomato plants after treatment with GA3 and looked at different aspects of the plant response, including Peroxidase, Superoxide dismutase and Catalase activities. Furthermore, a transcriptomics analysis was performed followed by a qPCR validation of the Illumina sequencing.
The work looks solid but there are several aspects that need to be reviewed and improved to have this manuscript ready.
Abstract:
Lane 20. Usually, transgenic plants are not referred as "strains" but as "lines". This comment is also present in lane 46 in the Introduction.
Introduction:
Lane 48. "isolated" is not the proper word here. Maybe.."whiteflies can be controlled...." would be better.
Changes in plant hormones's status have been reported multiple times in the literature in response to geminivirus infection (in tomato and in other plants). The authors should use that previous literature to enhance the introduction.
Lane 83. The authors state that "Most TYLCV strains are transmitted by whiteflies". This reviewer is not aware of TYLCV strains that are naturally not transmitted by whiteflies. Please clarify or remove.
Results:
Lane 103. It is not clear if the authors created a standard curve to measure viral loads. Furthermore, the authors relied on a single peak shown in the qPCR to claim the presence of the virus but do not explain clearly why is that. Readers unfamiliar with the use of the dissociation curve may find difficult to follow this. It needs to be clarified.
Lane 138, the authors need to clarify that the TYLCV was introduced into the plant by using Agrobacterium tumefaciens, and they did not just inoculated the bacteria. Clarify for the reader that the bacteria is used as a vector in this case.
Figure 3. In my version of the manuscript, I can't distinguish anything clear in the figure, the images are very poor, either remove them or make them look better.
Lane 162. This should move to the discussion.
Figure 6. this figure can't be read. There must be another way of portraying these results that can be visualized better, or simply move it to the supplementary data where you can also create excel sheets with lists that the interested reader can go and dig information from it. Maybe reducing the number of columns shown to a few highlighted groups would be better.
Figure 7. Similarly to figure 6, it may not be relevant to have them in the main manuscript. The suggestion to move them to the supplemental information is only that, a suggestion.
Materials and Methods:
Lane 398. Please clarify if the inoculation method used a needle with the syringe or not, some researchers infiltrate and others use the needle. Clarify the method used.
Discussion:
This reviewer considers that there is a big analysis missed from the manuscript. The authors didn't compare directly their findings with those of other researchers that have looked at the transcriptomic response to TYLCV in tomato (and other systems). The authors must execute that comparison (side by side) so they can assess what is different between a regular infection and the infection+GA3, this will narrow down the number of genes and pathways that are directly related to the question the authors are trying to answer. And will provide with a focus to proceed with the research. Without this analysis, the value of the manuscript is not as high as it could. The authors have all the information already to perform this analysis and no further experiments need to be done for it, just some time invested in looking closely to the data.
Comments on the Quality of English LanguageThere are some issues with the English, particularly some words (see comments in the previous section) but also writing "following the reported methods" in Materials and Methods seems overused and in other cases, the authors describe the method anyway, please be consistent.
Reviewer 3 Report
Comments and Suggestions for Authors
This manuscript titled “Gibberellin positively regulates tomato’s resistance to Tomato 2 Yellow Leaf Curl Virus (TYLCV) “ by Zhang et al. showed that 1) GA induced resistance to TYLCV in tomato inoculated using Agrobacterium carrying infectious cDNA of TYLCV, 2) TYLCV infection enhanced CA3 expression, 3) GA spray alleviated virus symptoms and reduced the virus accumulation, enhanced the activity of antioxidant enzymes and lowered reactive oxygen levels in tomato leaves, 4) GA3 upregulated genes associated with disease resistance, such as WRKYs, NACs, while downregulating DELLA protein. The content is simple and easy to understand. However, the manuscript is not well written with flaws, mistakes, corrections, etc. Besides those, the manuscript should be checked by a native English expert in this field. From these points this is not acceptable for this Journal in the present form. Some comments are described below.
1)Title: Gibberellin positively regulates tomato’s resistance to Tomato 2 Yellow Leaf Curl Virus (TYLCV) needs English improvements. It would be better to use (in) tomato instead of tomato’s. The same thing is in line 93. Tomato Yellow Leaf Curl Virus can be Tomato leaf curl virus just like one in key words.
2)Line 43: Viruses infiltrate plants needs English improvement. Infiltrate is not appropriate in this context.
3)Line 46: the use of resistant plant strains; strains are not appropriate. It needs improvement.
4)Line 58: When viruses replicate in large quantities; this is confusing for most readers because viruses means usually viruses of different species. It needs English improvement.
5)Line 118: TZR is the first appearance here, not in line 401. Describe the full name followed by abbreviated name in parenthesis. More explanation needed here or in other parts in the same section and/or discussion, pointing out that TRZ is unique in terms of response after the treatment, contrasted in most of other phytohormones.
6)Lines 261-263: Carrot ABA-induced in somatic embryos 3-like, Mitogen-activated protein kinase 261 kinase kinase 18 (MAPKKK18)-like, Pathogenesis-related protein PR-5 precursor, Wound- induced proteinase inhibitor 1 precursor were significantly up-regulated by GA3, while the same terms are all in lower case letters at their first letters in Table 3. These should be universally used in the text.
7)Fig.3: A; Vertical axis could be better to use “virus accumulation” than “relative level”. B: More clear pictures are needed. Those are difficult to differentiate one another. The plant growth by GA3 seems to be lower than those by H2O and P333, especially at 25 days. It may better to show any bar graph of growth retardation among three treatments in fig. or as supplementary data.
8)Fig. 4: There is neither explanation on DAB staining in details nor in Materials and Methods. Describe these.
Comments on the Quality of English Language
Needs improvements in many locations.
Round 2
Reviewer 3 Report
Comments and Suggestions for Authors
This is the revised manuscript which has been considered based the comments by this reviewer. However there are still some points which need to be modified. Some comments are described below.
1)Title: Tomato Yellow Leaf Curl Virus needs to change to Tomato yellow leaf curl virus. It should be just like one in key words.
2)Line 43: Viruses infiltrate plants needs English improvement. Infiltrate is not appropriate in this context.
3)Line 47: Instead of line, varieties would be more appropriate.
4)Line 61: the Mal de Rio Cuarto virus (MRCV); no “the” is needed. Rio should be Río. Be careful.
5)Fig. 1: A and C: Numbers in both axis are too small to see。These should be in the same letter size as in B.
6)Fig.3:. B: There are no mock inoculated control plants without symptoms. More clear pictures of H2O are needed compared with those of GA3 and PP333. It may better to show any bar graph of growth retardation among three treatments in fig. or as supplementary data.
7)Fig. 5: The letter sizes in both GA3 and PP333 figs are too small to see. These should be similar to those of Fig. 4.
8)Fig. 7: B; KEGG Enrichment-PP333-Down, Letter sizes should be the same as those of KEGG Enrichment-PP333-Up. There are different letter size between the two.

Comments on the Quality of English LanguageNeeds vigorous improvements.
Round 3
Reviewer 3 Report
Comments and Suggestions for Authors
This is the 2nd revised manuscript which has been considered substantially. However, there are still many corrections needed especially in references. Some comments are described below.
1)Fig.3: There are no mock inoculated control healthy plants. No answer comes from the authors. It will contribute to more understanding the data.
2)References: Many corrections are needed such as capital and lower case letters, italic/non italic etc..
Ref #3: the names of the viruses should be Tomato yellow leaf curl virus, Tomato chlorosis virus as shown in the cited paper..
Ref #5: the virus name should be Sugarcane mosaic virus.
Ref #7: Arabidopsis should be in italic.
Ref #14; plum pox virus and prunus should be Plum pox virus
and Prunus, respectively.
Ref #15: potato virus Y should be Potato virus Y in italic.
In this way many corrections are needed in other references.
Comments on the Quality of English Language
It needs to be checked by a native English expert in the similar fields.
